# Urinary Neuropilin-1: A Predictive Biomarker for Renal Outcome in Lupus Nephritis

**DOI:** 10.3390/ijms20184601

**Published:** 2019-09-17

**Authors:** Maria Teresa Torres-Salido, Mireia Sanchis, Cristina Solé, Teresa Moliné, Marta Vidal, Xavier Vidal, Anna Solà, Georgina Hotter, Josep Ordi-Ros, Josefina Cortés-Hernández

**Affiliations:** 1Hospital Quironsalud del Vallés, Internal Medicine Department, 08202 Sabadell, Spain; belovedmaite@hotmail.com; 2Hospital Universitari Vall d’Hebron, Vall d’Hebron Research Institute (VHIR), Rheumatology research group, Lupus Unit, 08035 Barcelona, Spain; msanchis@vhebron.net (M.S.); jordi@vhebron.net (J.O.-R.); fina.cortes@vhir.org (J.C.-H.); 3Hospital Universitari Vall d’Hebron, Department of Renal Pathology, 08035 Barcelona, Spain; teresa.moline@vhir.org (T.M.); mvidal@tauli.cat (M.V.); 4Clinical Pharmacology Service, Department of Pharmacology, Therapeutics and Toxicology, Fundació Institut Català de Farmacologia, Hospital Universitari Vall d’Hebron, Universitat Autònoma de Barcelona, 08035 Barcelona, Spain; xvg@icf.uab.es; 5Department of Experimental Pathology, IIBB-CSIC-IDIBAPS, 08036 Barcelona, Spain; anna.sola@iibb.csic.es (A.S.); georgina.hotter@iibb.csic.es (G.H.); 6CIBER-BBN, Networking Centre on Bioengineering, Biomaterials and Nanomedicine (CIBER-BBN), 08036 Barcelona, Spain

**Keywords:** neuropilin-1, lupus nephritis, urinary biomarker, clinical responder, renal biopsy

## Abstract

At present, Lupus Nephritis (LN) is still awaiting a biomarker to better monitor disease activity, guide clinical treatment, and predict a patient’s long-term outcome. In the last decade, novel biomarkers have been identified to monitor the disease, but none have been incorporated into clinical practice. The transmembrane receptor neuropilin-1 (NRP-1) is highly expressed by mesangial cells and its genetic deletion results in proteinuric disease and glomerulosclerosis. NRP-1 is increased in kidney biopsies of LN. In this work we were interested in determining whether urinary NRP-1 levels could be a biomarker of clinical response in LN. Our results show that patients with active LN have increased levels of urinary NRP-1. When patients were divided according to clinical response, responders displayed higher urinary and tissue NRP-1 levels at the time of renal biopsy. Areas under the receiver operating characteristic curve, comparing baseline creatinine, proteinuria, urinary NRP-1, and VEGFA protein levels, showed NRP-1 to be an independent predictor for clinical response. In addition, in vitro studies suggest that NRP-1could promote renal recovery through endothelial proliferation and migration, mesangial migration and local T cell cytotoxicity. Based on these results, NRP-1 may be used as an early prognostic biomarker in LN.

## 1. Introduction

Systemic lupus erythematosus (SLE) is a chronic inflammatory autoimmune disease with a broad spectrum of manifestations and organ involvement [1,2]. Lupus nephritis (LN) affects up to 50% of patients with SLE and is a major cause of morbidity, despite modern therapeutic approaches [2,3]. To date, renal biopsy is still the gold standard for diagnosing and classifying the degree of renal inflammation and scarring, but its invasiveness makes it unsuitable for serial monitoring. Conventional clinical parameters are not sensitive or specific enough for detecting ongoing disease activity, early relapse, disease progression, or response to therapy [2,3,4]. In recent years there has been increasing interest in developing novel biomarkers; however, few of them have been rigorously validated in large scale longitudinal studies, none have been standardized for daily clinical practice, and mainly they had been assessed for their potential as “diagnostic biomarkers”.

Neuropilin-1 (NRP-1) was originally identified as co-receptor for class 3 semaphorins, a family of molecules involved in axon repulsion [5]. Besides its critical role during embryogenesis, NRP-1 has important functions in adult tissues, being involved in axonal guidance, vascular endothelial sprouting, regeneration organ repair and immunosuppression. Evidence suggests that its role as immune-regulator would be in mediating interactions between activated dendritic cells and resting T cells [6,7,8,9,10,11,12,13], and promoting regulatory T cell activity [11].

Although originally detected in neurons, NRP-1 is also expressed in non-neural cells including epithelial and endothelial cells, where it also acts as a receptor for several members of the vascular endothelial growth factor (VEGF) family, enhancing signaling and promoting angiogenesis [14,15,16]. Other ligands include hepatocyte growth factor, platelet-derived growth factor BB, transforming growth factor-β1 (TGF-β1), and fibroblast growth factor2 (FGF2) [17,18]. In addition, it has been found that NRP-1 is present on cultured human mesangial cells and with the presence of VEGF play an important role in mesangial cell pathophysiology [19].

Little is known about the involvement of NRP-1 in SLE, and its role in lupus nephritis. Decreased NRP-1 and SEMA3A expression on serum B cells from patients with SLE has been described [20,21]. Vasdaz et al. in a study including 12 patients with LN showed an increased renal expression of NRP-1 in deposits found only in the damaged glomerular areas correlated with proteinuria and chronicity index (endothelial damage) [22]. So far, there are no studies evaluating NRP-1 as a biomarker of LN. Our aim was to evaluate expression levels of NRP-1 at the time of the renal biopsy in patients with LN and determine their predictive value for nephritis therapy response.

## 2. Results

### 2.1. Patients

A total of seventy patients with active LN were included in the cross-sectional study. Patients’ demographic characteristics and laboratory measurements are summarized in Table 1. Mean (SD) age at inclusion was 41 (±10) years. There was a female (80%) and biopsy-proven type IV glomerulonephritis (GMN) (76%) predominance. For 41 (58.6%) of patients, it was the first renal episode. The mean (SD) of renal activity and chronicity indices were 7.5 (3.6) and 2 (1.9), respectively. Patients with active nephritis were treated with standard therapy (see material and methods) and were followed for a median of 6 years (range 1.5 to 8.5 years).

### 2.2. Urinary Expression of NRP-1 in Lupus Nephritis

Urinary NRP-1 mRNA levels in patients with active LN were significantly increased when compared with other groups: SLE patients with active disease without renal involvement (30.70 ± 111.8 vs. 1.03 ± 0.94 relative expression; *p* = 0.01), other glomerular diseases (3.35 ± 3.65 relative expression; *p* = 0.03), and healthy controls (1.03 ± 0.95 relative expression; *p* = 0.02) (Figure 1A). Among patients with active LN, there were no differences in NRP-1 levels according to histological type or the degree of disease activity.

### 2.3. Protein Levels of NRP-1 in Patients with Lupus Nephritis

In line with mRNA expression levels, patients with active LN had significantly higher urinary NRP-1 levels, as measured by ELISA, than lupus patients with active non-renal disease (1807 ± 2180 ng/mg Cr versus 95.26 ± 160.3 ng/mg Cr, *p* < 0.0001), patients with other glomerular diseases (1807 ± 2180 ng/mg Cr versus 13.11 ± 17.77 ng/mg Cr, *p* < 0.0001) and healthy controls (1807 ± 2180 ng/mg Cr versus 59.14 ± 26.39 ng/mg Cr, *p* < 0.0001) (Figure 1B). Levels of uNRP-1 (area under curve = 0.956) showed a sensitivity of 85.71% and specificity of 90.24% to discriminate active nephritis from active non-renal lupus disease (cut-off >293.55, Appendix A). Correlation analysis of uNRP-1 levels with other clinical and laboratory parameters only showed a correlation with C4 levels (*r* = −0.182, *p* = 0.045, Figure 1C). Higher urine levels of NRP-1 did not correlate with type IV glomerulonephritis or the degree of proteinuria, biopsy activity index, or urinary sediment. However, an inverse correlation with nephritic flare was found (*r* = −0.246, *p* = 0.043, Figure 1C). Measurement of serum NRP-1 levels did not show differences between groups (Appendix A).

### 2.4. Baseline Levels of NRP-1 Predict Clinical Response

Of the 70 included patients, 38 (54%) achieved complete response within a median of 18.5 months (range: 2 to 65). Table 2 shows the baseline characteristics between responders and non-responders. Non-responders had significantly higher levels of serum creatinine, more nephritic flares and relapsing disease when compared with responders (*p* = 0.02, *p* < 0.0001, and *p* = 0.02, respectively). Measurement of NRP-1 at inclusion showed responders to have higher urinary mRNA levels of NRP-1 than non-responders (102 ± 210 vs. 8 ± 18 relative expression, *p* = 0.002) (Figure 1C). Differences were more significant at a protein level (2532 ± 2439 ng/mg Cr vs. 569 ± 851 ng/ mg Cr, *p* < 0.0001) (Figure 1D).

Next, we performed receiver operating characteristic analyses to determine the best cutoff that had the maximal sensitivity and specificity to predict clinical response. A urinary NRP-1 cutoff level of 1143 ng/mg Cr at baseline could predict clinical response with a sensitivity of 87.10%, specificity of 71.79% and a negative and positive predictive value of 87.50% and 71.05%, respectively (Figure 2A). Patients with higher uNRP1 baseline levels had a significantly increased probability to achieve CR (*p* < 0.0001, by log-rank test) (Figure 2B).

### 2.5. VEGFA, VEGFR1, VEGFR2, and SEMA3A in Patients with Lupus Nephritis and Correlation with NRP-1 Levels

As there is a functional relationship between NRP-1 and the VEGF and the semaphorine family, we measured the urinary mRNA and protein levels of VEGFA, VEGFR1, VEGFR2, and semaphorine3A (SEMA3A). Urinary gene expression levels did not differ between patients with active LN when compared with control groups, nor between responders and non-responders. However, urinary VEGFA levels, measured by ELISA, were significantly increased in patients with active LN compared with active SLE patients without renal disease and healthy controls (*p* = 0.01 and *p* < 0.0001, respectively) (Appendix A). In contrast, urinary SEMA3A levels were significantly decreased in patients with active LN when compared to the same control groups (*p* = 0.002 and *p* = 0.007, respectively) (Appendix A). There were no differences in urinary VEGFR1 and VEGFR2 levels (Appendix A). When responders were compared with non-responders, mirroring NRP-1 levels, responders had significantly higher levels of urinary VEGFA compared with non-responders (166 ± 9 vs. 87 ± 7 ng/mg Cr, *p* < 0.0001) (Figure 2C). No differences in urinary VEGF receptors and SEMA3A levels were found between responders and non-responders (Appendix A).

There was a direct correlation between urinary VEGFA and NRP-1 protein levels (*r* = 0.466, *p* < 0.0001, Figure 2D). However, VEGFA levels (12.34 ng/mg) showed an inferior profile to predict response with a sensitivity and specificity of 71% and 75%, respectively (Appendix A). ROC analysis comparing NRP-1 with baseline urinary VEGFA levels and other clinical factors such as creatinine and proteinuria levels as predictors of response continued to show uNRP-1 levels to be an independent predictive factor of response (*p* = 0.003) (Figure 2E).

### 2.6. Immunohistochemistry Localization of NRP-1, VEGFA and SEMA3A within Renal Biopsies

NRP-1 and VEGFA stained with a significantly higher intensity in the active LN group when compared to the control group (Figure 3A). NRP-1 expression was mostly observed in the glomeruli and along the tubuli. In the glomeruli, it was mostly expressed in the endothelial cell membrane and in mesangial cells. Globally, NRP-1 expression was three times higher in responders than in non-responders. There was a clear difference in the glomeruli staining, with responders having much higher expression of NRP-1 (*p* < 0.0005). This difference was not observed in the tubuli (Figure 3A). Renal NRP-1 protein levels correlated with urinary levels (*p* = 0.002, *r* = 0.834). VEGFA was highly expressed in the glomeruli and tubular epithelial cells, but staining intensity did not differ between responders and non-responders. Finally, semaphorine 3A staining was increased in patients with active LN when compared to controls and expressed strongly in the tubuli of non-responders (Figure 3A).

### 2.7. Serial Monitoring of uNRP-1 Levels to Predict Response

A new prospective cohort (*n* = 39) (cohort 3) (22 responders and 17 non-responders) was established. Serial measurements of urine NRP-1 and VEGFA levels were performed three-monthly from baseline up to twelve months (Table 3). During follow-up, NRP-1 levels remained markedly increased in the responder group (median levels 1112 ng/mg Cr, *p* = 0.003), when compared with non-responders (median levels 149.1 ng/mg, Figure 3B). VEGFA levels during the first 6 months of follow-up were significantly increased when compared to non-responders (levels 303 vs. 88 pg/mg, *p* = 0.05), but afterwards, levels remained similar in both study groups (around 100–80 pg/mg, Figure 3C).

### 2.8. NRP-1 Expression Levels by Primary Cells under Stimulatory Conditions

To establish a link between inflammation and NRP-1 production, primary renal cells were stimulated by inflammatory cytokines such as IL1-α and VEGF. At 24 h, following IL1-α stimulation, NRP-1 expression levels were significantly increased in mesangial renal cells when compared to the other cellular types (*p* < 0.0001). However, following VEGF stimulation, endothelial renal cells exhibited the highest expression levels of NRP-1. Under stimulation, T cells reduced significatively the NRP-1 expression (Figure 4A). 

### 2.9. Effect of NRP-1 on Endothelial Proliferation and Migration

To further investigate the role of NRP-1 in LN and its contribution to renal recovery by enhancing angiogenesis, we inhibited NRP-1 in primary human renal endothelial cell (HREC) and performed proliferation, migration and wound-healing assays. A significant downregulation of HREC cell proliferation was observed in NRP-1 inhibited HREC cells regardless of VEGF protein presence (Figure 4B). In addition, no further HREC cell migration was observed beyond 24 h (*p* < 0.001, Figure 4C). VEGFR2 and PI3K expression levels were significantly upregulated in inhibited NRP-1 HERC following VEGF protein stimulation (5.8 and 3.3 fold change, respectively, Figure 4D). However, low levels of NRP-1 and VEGFR2 protein were observed in NRP-1 inhibited HREC cells by immunofluorescence (Figure 4E). Data suggest NRP-1 angiogenesis effect to be through a VEGF-dependent pathway.

### 2.10. NRP-1 Induces Migration in Human Renal Mesangial Cells Via PDGFB

Production of inflammatory cytokines and proliferation assays were evaluated in inhibited NRP-1 human renal mesangial cells (HRMCs). Following VEGF stimulation, no significant differences in IL6, IL8, and CCL2 gene expression were observed between inhibited NRP-1 HRMCs and controls (Appendix A). Neither changes in proliferation were observed (Appendix A). NRP-1 is a versatile transmembrane coreceptor for VEGFA and also of PDGFB, known for its chemotactic effect on HRMC. Therefore, we investigated whether NRP-1 is required for HRMC migration using in vitro Transwell migration assay following PDGFB stimulation (Appendix A). A significant reduction of relative cell migration was observed in inhibited NRP-1 HRMCs following PDGFB stimulation (fold migration of −3.11, *p* < 0.001, Figure 5A). This reduction was not observed in non-stimulatory conditions (Figure 6. So, under PDGFB stimulation, NRP-1 gene expression is necessary to induce HRMC migration.

### 2.11. NRP-1 Induces a VEGF-Dependent T cell Cytotoxicity

VEGF directly suppresses T cell activation via VEGFR2, and NRP-1 is a coreceptor of VEGFR2. So, NRP-1 may have an immune suppression effect in T cells localized in LN kidneys. We inhibited the NRP-1 gene expression from peripheral T cells isolated from patients with LN. Following VEGF stimulation, no significant difference was observed in T cell activation (data not shown). However, VEGF induced a significant cytotoxicity in control CD4^+^ T cells and CD8^+^ T cells not observed in inhibited NRP-1 T cells (48–58% vs 0.1–0.2% death cells, respectively, *p* < 0.0001, Figure 5B). Data suggest significantly increased VEGF-dependent T cell death in the presence of NRP-1.

## 3. Discussion

Attainment of complete remission in lupus nephritis is associated with significantly improved prognosis and flare rate reduction [23]. Despite improvements in early diagnosis, close monitoring, and intensive treatment, only a small proportion of patients will achieve remission. Several clinical and histological factors have been associated with bad prognosis and progression to ESRD [24,25]. However, to date, current biomarkers are not reliable in predicting treatment response. The present study shows NRP-1 as a biomarker of active LN and verifies its potential utility as an early biomarker of therapeutic response.

This is the first study showing urinary NRP-1 both at a protein and gene expression level to be significantly increased in patients with active lupus nephritis. NRP-1 urinary levels could discriminate active renal disease from non-active disease with a sensitivity and specificity of 73.33% and 97.5%, respectively, suggesting that its measurement is useful in diagnosis and a reliable marker of renal disease activity in patients with SLE. NRP-1 deposits were also increased in the glomeruli of patients with active renal disease when compared with normal kidneys and correlated with urinary NRP-1 levels. Vasdaz et al. had previously reported high levels of NRP-1 in the renal tissue of patients with LN [22]. The study showed a positive correlation between NRP-1 levels and the clinic-pathological parameters of renal disease. In addition, they observed a higher intensity of staining in patients with focal GMN compared with those with diffuse GN, suggesting that NRP-1 could serve as a biomarker to distinguish both types of glomerulonephritis [22]. Unlike that study, we could not find differences between the different histological types of LN but our study was not powered to see differences between the subgroups. In view of the relationship between NRP-1, VEGF and the semaphorine family, VEGF and semaphorine 3A levels were also analyzed in patients with active LN. In agreement with Vasdaz et al. [22], we found that patients with active LN had increased VEGF levels in urine and renal tissue that correlated with NRP-1 levels. Like in their study, semaphorine 3A staining was increased in active LN and observed predominantly in the renal tubuli but in our study urinary levels were significantly reduced in patients with active disease There is no data available on levels of semaphorine 3A in lupus nephritis to compare. Studies on serum and plasma had shown semaphorine 3A levels to be decreased in patients with lupus disease compared to controls. Levels were found to be inversely correlated with disease activity, mainly in the presence of lupus nephritis [20,21].

Baseline levels were also found to be a useful predictor of clinical response following therapy. Responders had significantly higher urinary NRP-1 levels as well as during follow-up when compared with non-responders. NRP-1 renal expression was also increased in responder patients and found predominantly in the glomeruli unlike the non-responders. Mirroring NRP-1 levels, responders had also significantly increased VEGF urinary levels but no differences in renal staining were found between the groups. Conversely, responders showed a decreased semaphorine 3A renal staining compared to non-responders in which semaphorine 3A was strongly expressed in the tubuli. Previous studies have already suggested a role for Semaphorine 3A in renal damage [22]. There were no differences in urinary semaphorine 3A levels. After analyzing the predictive value for CR amongst several predictive factors, urinary NRP-1 levels at the time of renal biopsy were found to be an independent factor of clinical response and responders had persistently increased levels during follow-up. Whereas this is the first time that the prognosis value of NRP-1 is shown, the prognosis value of VEGF in LN has been previously reported [26]. Decreased renal expression was associated with worse prognosis, serving as a molecular marker of renal damage and a short-term loss of kidney function in those patients.

To date, little is known concerning the involvement of NRP-1 in SLE, and even less about its role in LN. There are still some open questions, whether this molecule is pathogenic or to some extent protective. Although the exact mechanism is not well known, our data support that NRP-1 along with VEGF and semaphorine are proteins involved in LN and have an important role in renal recovery.

NRP-1 serves as a receptor for VEGF. Several studies have suggested that VEGF plays a key protective role by maintaining normal glomerular homeostasis and the integrity of glomerular endothelial cells [15,19]. Our in vitro studies show a link between inflammation and production of NRP-1 by the primary renal cells. The production is clearly influenced by the predominant inflammatory cytokine in the environment. So, following IL1 stimulation, NRP-1 was mainly produced by mesangial cells, whereas following VEGF, endothelial cells were the one showing stronger expression levels of NRP-1. Since high levels of NRP-1 were found in therapeutic responders, it suggests a role for this molecule in renal repair. The in-vitro studies describe different mechanisms by which NRP-1 may have this role. Our study demonstrates that NRP-1 enhances VEGF activity, intensifying its protective effect by increasing its angiogenic activity (Figure 6). Proliferation of new blood vessels and consequently a greater oxygen supply in the glomerular endothelium provide the environment to enhance renal recovery [27]. On the other hand, we observed that inhibition of NRP-1 decreased migration of human mesangial renal cells in a PDGFB-dependent pathway. This indicates that NRP-1 is required for cell motility signaling driven by PDGFB. Migration of mesangial cells is important in repair and maintenance of the mature glomerulus [28,29]. Finally, VEGF suppresses activation of T cells via VEGFR2 [30,31]. In our work, we demonstrated that in inhibited NRP-1 cells, VEGF stimulation did not provoke T cell death. Conversely, VEGF promoted T cell death in the presence of NRP-1. A high amount of CD8^+^ T cells have been shown in the renal biopsies of patients with LN in comparison with other T cell subsets such as CD4^+^ or Treg cells [32]. NRP1 receptor has been described in CD8^+^ T cells [33]. However, we did not see differences in the dead cellular rate between CD4^+^ and CD8^+^. In view of these results, we hypothesized that high levels of NRP-1 in the presence of VEGF could increase T-cell death, which in turn could decrease the inflammatory and cytotoxicity renal state as a first step to renal recovery (Figure 6).

The main limitations of the study are its single-center nature and the relatively short period of follow-up in the longitudinal study. Validation in a larger prospective cohort is therefore necessary. Further investigation of the role of NRP-1 in animal models will help to understand its role in renal repair.

In conclusion, our results demonstrate that urinary and renal NRP-1 levels at the time of renal flare may serve as a novel prognostic biomarker for prediction of clinical response in lupus nephritis.

## 4. Materials and Methods

### 4.1. Patients

We established an initial exploratory cross-sectional cohort (*n* = 25) followed by a validation cohort (*n* = 45). For the purpose of the study both cohorts were combined and presented as the primary cohort of patients with LN (*n* = 70). Details of the two different cohorts are shown in SI (Appendix A). In addition, a new prospective cohort of patients with active LN (*n* = 39) was also established. Active lupus nephritis was defined by either a total urinary protein level ≥0.5 g/day, an increment of serum creatinine levels of more than 0.5 mg/dL or the presence of active sediment by microscopic examination, and confirmed by renal biopsy. All patients fulfilled at least 4 of the American College of Rheumatology (ACR) revised classification criteria for SLE [34]. SLE disease activity was assessed by the SLE Disease Activity Index 2000 update (SLEDAI-2Ks; range 0–105) [35]. Renal SLEDAI refers to the sum of the SLEDAI-2Ks accrued in the renal domain of the measure (rSLEDAIs; range 0–16). All patients were treated with IV methyl-prednisolone (500 mgx3) followed by a tapering dose of oral prednisone along with at least a 24-month course of oral mycophenolate mofetil. Results were also analyzed according to clinical outcome after treatment (responders (*n* = 38) vs. non-responders (*n* = 32). Following therapy, complete response (CR) was defined as urinary protein excretion <0.2 g/24 h, normal urinary sediment and normal or stable renal function (within 10% of normal eGFR if previously abnormal renal function). No clinical response was defined by a urinary protein excretion ≥0.2 g/24 h, increase in the serum creatinine concentration ≥0.6 mg/dL, a creatinine clearance >15% below the baseline value discontinuation of treatment due to side effects or renal relapse while on full treatment [36].

More information about renal flare [37] kidney biopsy evaluation [24,38] and, treatment [39] in Supporting Information. 

Urine and blood samples were collected from each patient 1 day before renal biopsy and processed immediately to be stored at −80 °C. Patients with urinary tract infection, diabetes mellitus, pregnancy, malignancy, and non-lupus-related renal failure were not included.

The control groups consisted of SLE patients with active disease (SLEDAI > 6) without renal involvement (*n* = 25), healthy volunteers (*n* = 25) and patients with other glomerular diseases (*n* = 25) (diabetic nephropathy (*n* = 12), IgA nephropathy (*n* = 4), idiopathic membranous nephropathy (*n* = 4), and focal segmental glomerulosclerosis (*n* = 5) were also included.

The study was approved by the Vall d’Hebron Ethic Committee (Urinary Biomarkers LN, 22 April 2009) and written informed consent was obtained from all patients. More details about study design in Supporting Information.

### 4.2. mRNA Isolation, cDNA Synthesis and Real-Time PCR

Urine samples were immediately centrifuged after collection at 3900× *g* for 30 min (4 °C) and supernatant was stored at −80 °C. Total RNA was isolated from the cell pellets using the RiboPure RNA Purification Kit (Ambion, Life Technology, Carlsbad, CA, USA) following the manufacturer’s instructions. Quantification of NRP-1, VEGFA, VEGFR1, VEGFR2 and SEMA3A mRNA by TaqMan Real-Time PCR was carried out as described by the manufacturer (ThermoFisher, Carlsbad, CA, USA). RNA was reverse-transcribed into cDNA using the High Capacity RNA-to-cDNA Kit and the quantitative RT-PCR reaction was performed in 96-well plates on the ABI PRISM 7000HT. The data was normalized based on the expression of the endogenous control GAPDH (more details in Supporting Information).

### 4.3. Enzyme-Linked Immunosorbent Assay (ELISA)

Enzyme-linked immunosorbent assay (ELISA) was used to quantify urinary and serum levels of NRP-1 (Cloud-Cone Corp, Katy, TX, USA), VEGFA (RayBiotech, Norcross, GA, USA), VEGFRI (RayBiotech, Norcross, GA, USA), VEGFR2 (RayBiotech, Norcross, GA, USA), and SEMA3A (Cloud-Cone Corp, Katy, TX, USA) following the manufacturer’s instructions. Triplicate for all samples were done. Urinary levels were normalized against the corresponding urine creatinine levels.

### 4.4. Renal Histology and Immunohistochemistry

Paraffin sections were stained with haematoxylin and eosin, periodic acid-Schiff, trichrome, and silver for light microscopy. The specimens were scored for activity and chronicity [24]. All biopsies were examined by one pathologist that was not aware of the molecular study results. For immunohistochemistry, formalin-fixed paraffin-embedded tissue sections were deparaffinized and primary antibodies were incubated with slides overnight at 4 °C (more information in Supporting Information). As controls, we used disease-free kidney sections from tissue margins of total or subtotal nephrectomies obtained from patients undergoing surgery for renal malignancies.

### 4.5. Cell Cultures

Primary Human Renal Endothelial Cells (HRECs), Primary Human Renal Glomerular Mesangial cells (HRMCs) and Primary Human Renal Tubular Epithelial cells (RTECs) were purchased and cultured in the recommended media provided by the manufacturer (Innoprot, Derio, Bizkaia, Spain).

Isolated primary T cells were cultured in RPMI medium supplemented with 10% fetal bovine serum (FBS) and 5% penicillin/streptomycin (more details in Supporting Information). Cells were grown at 37 °C in a humidified 5% CO_2_ atmosphere. Cell passes were performed using TrypLE^TM^ Express (ThermoFisher, Carlsbad, CA, USA).

### 4.6. NRP-1 Expression Levels in Primary Cells

Primary human cells (HRECs, HRMCs, RTECs and T cells) were platted onto 24-wells (1 × 10^4^ cells/mL). After 24 h, stimulation with IL-1 (10 ng/mL) or VEGF (50 ng/mL) cytokine was performed during 24 h. PBS were added as control in non-stimulation conditions. After that, RNA extraction was performed to analyze NRP-1 expression levels by real-time PCR. 

### 4.7. NRP-1 Inhibiton and Stimulation Conditions

Primary human cells were cultured in their appropriate medium and at passage 2 were plated onto 24-well or 96-well Cell+ culture plates (Sarstedt, Nümbrecht, Germany). After 24 h, primary cells were confluence between 70–80% and they were transfected with gRNA NRP-1 or their negative-control using Lipofectamine^TM^ CRISPRMAX^TM^ Reagent Cas9 Nuclease Transfection Protocol (Invitrogen, Carlsbad, CA, USA) following the manufacturer’s instructions. After inhibition of NRP-1, renal cells were stimulated with different conditions during 24 h: 50 ng/mL of VEGF (Gibco® Thermofisher Scientific, Carlsbad, CA, USA), 50 ng/mL PDGFB (ThermoFisher, Carlsbad, CA, USA) or Non-stimulation as negative control. After that, wound healing assay, proliferation assay, apoptosis assay, immunofluorescence assay, or RNA extraction were performed.

### 4.8. Wound Healing Assay

To perform wound healing assays, we used a scratch assay in primary cultured cells inhibiting the NRP-1 gene and following stimulatory conditions. To calculate migration, we used the analysis of bright field microscopic images by determining the distance between wound edges over time compared to the control (more details in Supporting Information).

### 4.9. Proliferation Assay

After 24 h of stimulation, proliferation assays were performed using CyQUANT^®^ NF Cell Proliferation Assay Kit (ThermoFisher, Carlsbad, CA, USA) following the manufacturer’s instructions. The diluted CyQUANT^®^ NF dye reagent was added to every well and the fluorescence was measured using a fluorescence microplate reader with excitation at 485 nm and emission detection at 530 nm.

### 4.10. Migration Assay

Migration assay was conducted using Boyden chambers with 8-μm-pore polycarbonate membranes. The chambers were placed in culture wells in medium supplemented with PDGFB stimulation or non-stimulation. Cells that migrated across the chamber membrane within 6 h were stained with DAPI and quantified using immunofluorescence microscopy. At least, three independent experiments were performed, and results of representative experiments were shown.

### 4.11. Apoptosis Assay

Apoptosis assay were carried out using Dead Cell Apoptosis Kith with APC annexin V and SYTOX^®^ Green Flow Cytometry (ThermoFisher, Carlsbad, CA, USA) following the manufacturer’s instructions. Cells were stained using two fluorophores: Annexin V conjugated to allophycocyanin (APC) to detect apoptotic cell and SYTOX Green dye to stain dead cells. After incubation during 15 min at 37 °C, stained cells were analyzed by flow cytometry (BD LSRFortessa™, Haryana, India).

### 4.12. Immunofluorescence Assay

For immunofluorescence, cells were incubated with PFA 4% during 20 min. After washing with PBS, they were incubated with 0.1% triton during 10 min. Blocking using PBS 5% BSA was performed during 1 h at room temperature and primary antibody Rabbit pAb to VEGFR2 and NRP-1 Mouse monoclonal IgG (1:200, Santa Cruz Biotechnology, Dallas, TX, USA) were incubated overnight at 4 °C. Secondary antibodies were incubated during 2.5 h at room temperature (more details in Supporting Information).

### 4.13. Statistical Analysis

Raw threshold cycle (Ct) values were imported from ABI7000 SDS software and relative expression levels for each mRNA were calculated using the comparative Ct method. Continuous variables were compared using Student’s t-test or one-way ANOVA followed by Tukey’s multiple comparison test when appropriate. Categorical data were compared by the Chi-square test. Spearman’s rank-order correlation was used to analyze the correlation between two parameters. Time-to-event analyses were performed with Kaplan-Meier method and log-rank test. The diagnostic performance of biomarkers was evaluated by calculating their sensitivity and specificity using ROC curves. Cutoff values were determined according to Youden’s index. A *p* value <0.05 was considered statistically significant. Statistical analyses were performed by GraphPad Prism version 6 (GraphPad Software, La Jolla, CA, USA) and SAS version 9.3 (SAS Institute Inc., Cary, NC, USA). 

## Figures and Tables

**Figure 1 ijms-20-04601-f001:**
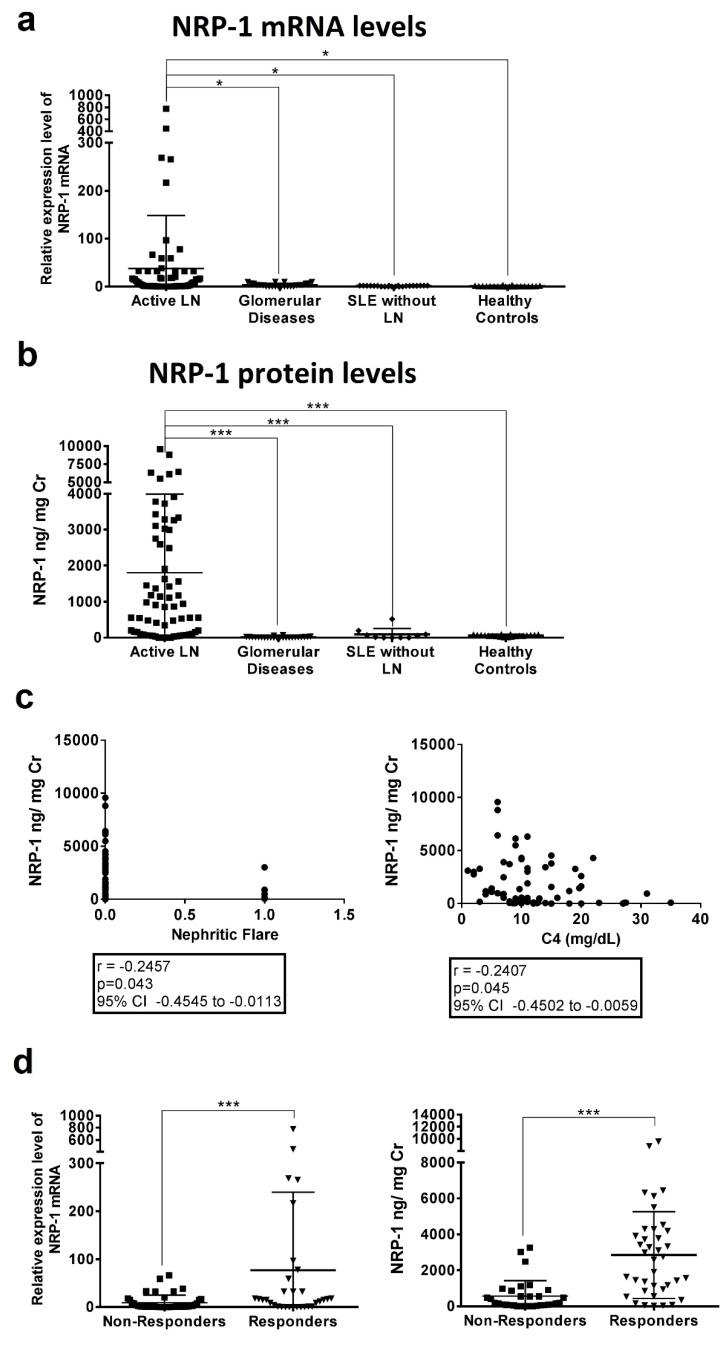
Urinary NRP-1 relative expression levels (**a**) and concentrations (**b**) in active LN patients compared with active SLE patients without renal disease, with other glomerular diseases and healthy controls. Urinary biomarker expression levels (**c**) and concentrations (**d**) were measured according to clinical response to therapy (responders vs non-responders). Biomarker concentrations were standardised to urinary creatinine and expressed as median values. Horizontal line means median value for each group. One-way ANOVA followed by Bonferroni test (**a**,**b**) and Student’s *t*-test (**d**) were used to to compare biomarker concentrations between groups. * *p* < 0.05; *** *p* < 0.0001.

**Figure 2 ijms-20-04601-f002:**
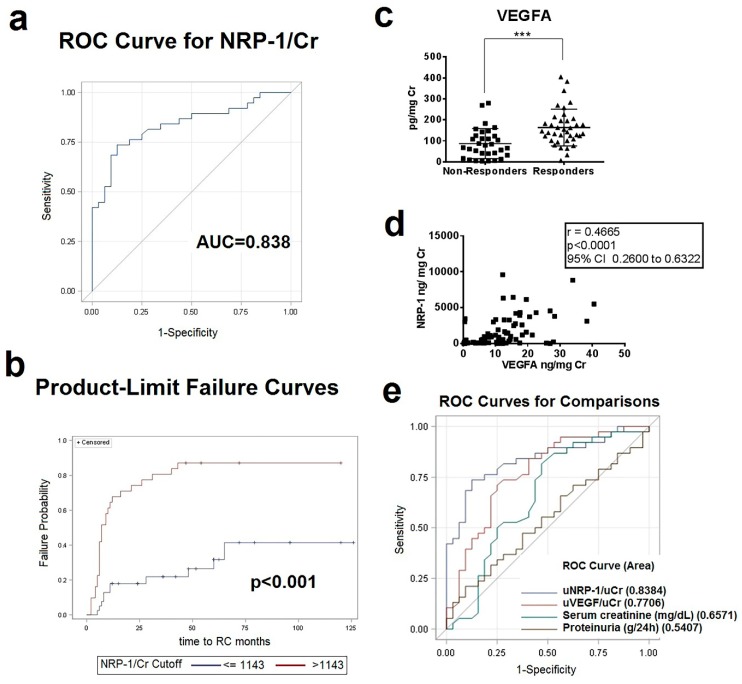
Urinary Levels of NRP-1 and vascular endothelial growth factor (VEGFA) in active LN. (**a**) The receiver operating characteristic (ROC) curve of urinary NRP-1 levels at the time of renal biopsy generated from the optimal binary logistic regression model when data from both cohorts were combined to discriminate between responders and non-responders. AUC = area under the ROC curve. (**b**) Kaplan-Meier survival curve for clinical response following treatment among patients with active LN at the time of renal biopsy. Long rank test was used for analysis. *p*-value ≤ 0.05 were considered significant. (**c**) Creatinine-normalized urine levels of VEGFA in combined cohorts of responders (*n* = 38) and non-responders (= 32); *** *p* < 0.0001. (**d**) Correlation plots of urinary NRP-1 and VEGFA levels at the time of renal biopsy. Creatinine-normalized urine levels of VEGFA in LN in relation to NRP-1 levels at the time of renal biopsy in patients with active LN. (**e**) Receiver operating characteristics curves relating the specificity and sensitivity profiles of baseline urinary NRP-1, VEGFA, creatinine and protein levels.

**Figure 3 ijms-20-04601-f003:**
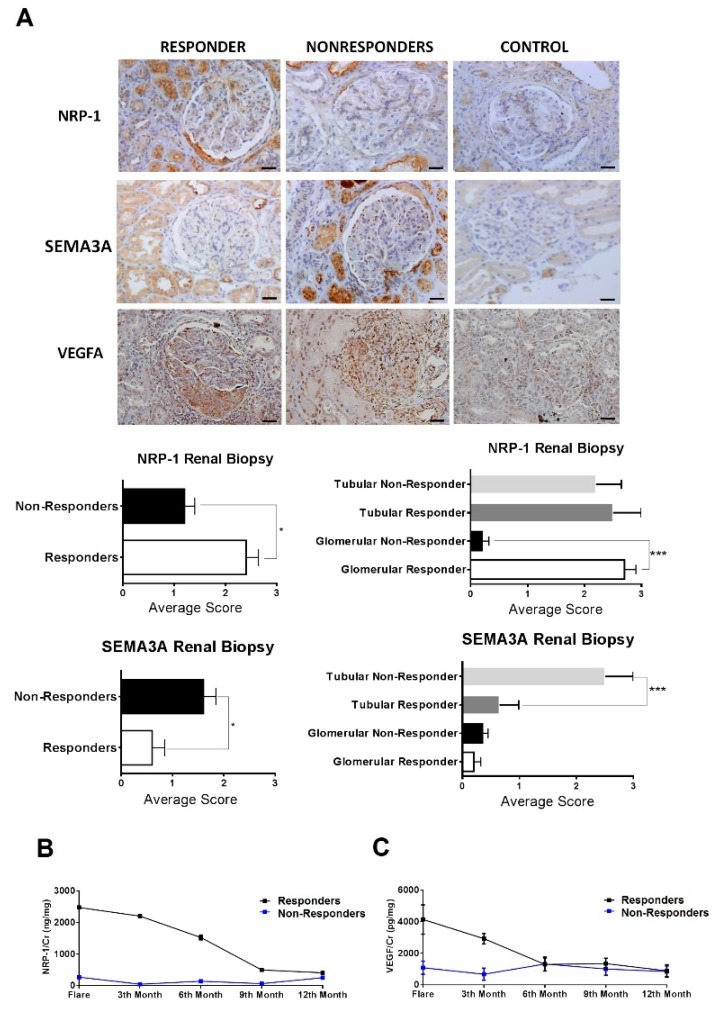
High levels of NRP-1 protein were confirmed in Responders vs Non-Responders. (**a**) Immunohistochemistry for NRP-1 and VEGFA in kidney biopsy (×400). Average scores were obtained for the evaluation of Vall Hebron pathologists group. (**b**) Changes in urinary protein levels of NRP-1 and VEGFA. (**c**) over the 12-month period of treatment in patients with LN. Dots and bars show means and SEM of protein levels in urine samples from baseline though the twelve months of follow-up. * *p* < 0.05.

**Figure 4 ijms-20-04601-f004:**
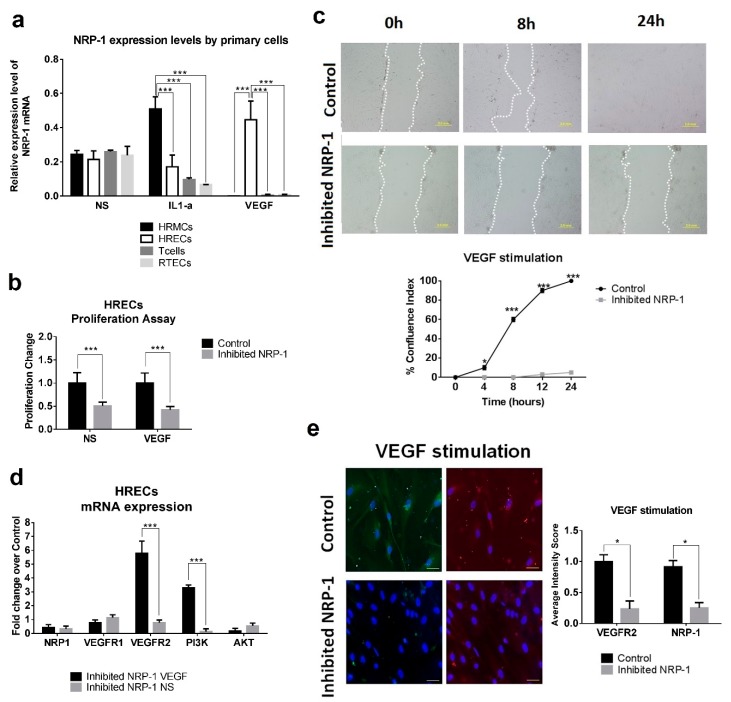
NRP-1 induced proliferation and migration in HRECs. (**a**) Expression levels of NRP-1 after non-stimulation (NS), interleukin-1 alpha (IL1-a) and vascular endothelial growth factor (VEGF) stimulation in primary human cells: renal mesangial cells (HRMCs), renal endothelial cells (HRECs), T cells and renal tubular epithelial cells (RTECs). One-way ANOVA (correction Bonferroni). *** *p* < 0.0001. (**b**) Proliferation assay in inhibited NRP-1 and control with VEGF stimulation or with non-stimulation conditions (NS). Student’s *t*-test. *** *p* < 0.0001 (**c**) A straight scratch was performed to simulate a wound. After that, the distance between wound edges was determinate overtime and in each condition. Migration rate was expressed as percentage of confluence index (CI) between the performed scratch and calculated using the distance between wound edges. Photographs were captured by a phase contrast microscope. Bars, 2 mm (**d**) Analysis of mRNA expression levels between inhibited NRP-1 HRECs in VEGF or NS conditions. Fold change was calculated over their transfection control. Student’s t-test. *** *p* < 0.0005 (**e**) Immunofluorescence of VEGFR2 (green) and NRP1 (red) in HRECs after transfection and stimulation. Bars, 50 µm. DAPI staining was used to label cell nuclei. Student’s *t*-test. * *p* < 0.05. HRECs: Human Renal Endothelial Cells.

**Figure 5 ijms-20-04601-f005:**
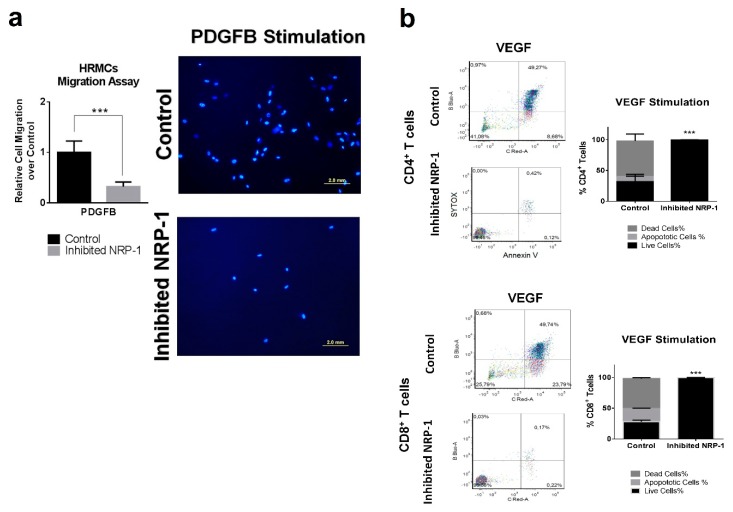
NRP-1 affects PDGFB migratory behavior in HRMCs and viability in T cells. (**a**) Migration assay of inhibited NRP-1 and control HRMCs in PDGFB stimulation conditions. DAPI staining was used to label migrated cells and they were counted using conventional fluorescence microscopy. Bars, 2.0 mm. Relative cell migration was calculated using control cells. *** *p* < 0.0005. (**b**) Flow cytometry were used to analysis the percentage of dead, apoptotic and live cells in primary human T cells after VEGF stimulation. In inhibited NRP-1 T cells, only live cells were observed. *** *p* < 0.0005.

**Figure 6 ijms-20-04601-f006:**
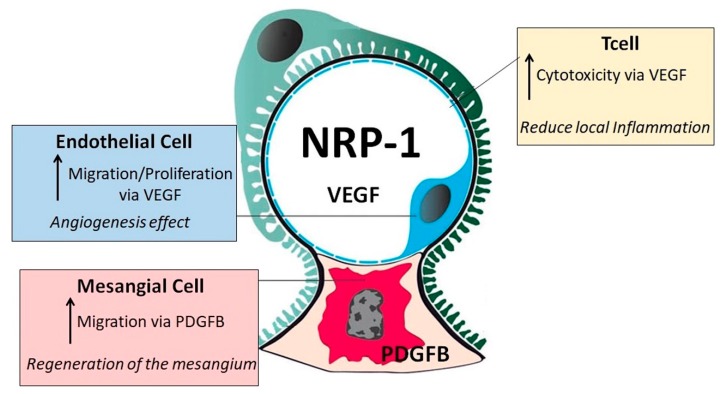
Proposed positive effect of NRP-1 in lupus nephritis renal recovery via increasing VEGF-angiogenesis effect in endothelial cells, PDGFB-regeneration of the mesangium and VEGF-lymphocyte cytotoxicity. Black arrow means “increase”.

**Table 1 ijms-20-04601-t001:** Clinical and histological variables at the time of the renal biopsy.

Characteristics	Active Lupus Nephritis (*n* = 70)	Active Non-Renal SLE (*n* = 25)	Other Glomerular Diseases (*n* = 25)	Healthy Controls (*n* = 25)	*p* Value ^a^
*Demographic*					
Age, yrs	41 ± 10	34 ± 6	35 ± 5	30 ± 4	0.4857 (0.787) [0.885]
Gender (Female/male)	56/14	21/4	16/9	16/9	0.112 (0.723) [0.112]
*Race/ethnicity, n (%)*					
White Hispanic	65 (93) 4 (6)	22 (88) 2 (8)	23 (92) 1 (4)	22 (88) 2 (8)	0.966 (0.966) [0.500]
Other	1 (1)	1 (4)	1 (4)	1 (4)	
*Laboratory parameters*					
Serum creatinine, mg/dL	1.1 ± 0.5	0.8 ± 0.1	1.9 ± 0.8	0.8 ± 0.3	0.035 (0.006) [0.001]
eGFR (mL/min)	82 ± 30	104 ± 16	48 ± 36	95 ± 22	0.091 (0.001) [0.006]
Urea (mg/dL)	54 ± 29	34 ± 14	81 ± 48	28 ± 11	0.076 (0.152) [0.012]
Anti-dsDNA Abs, IU/mL	287 ± 68	39 ± 29	n.a.	n.a.	(0.087)
Serum C3, mg/dL	78 ± 31	94 ± 20	n.a.	n.a.	(0.101)
Serum C4, mg/dL	12.8 ± 9.2	13.4 ± 10	n.a.	n.a.	(0.090)
Proteinuria, g/24 h	3.3 ± 3.0	0.2± 0.1	9,1 ± 4,8	n.a.	(0.010) [0.001]
Leukocytes (cel/µL)	95 ± 114	n.a.	n.a.	n.a.	
Erythrocytes (cel/µL)	133 ± 131	n.a.	n.a.	n.a.	
*Disease index (SLEDAI-2K)*					
Total score	16 ± 2	8 ± 2	n.a.	n.a.	(0.006)
Renal score	12 ± 1	0	n.a.	n.a.	
Extra-Renal score	4 ± 2	8 ± 2	n.a.	n.a.	(0.001)
*Renal Flare, n (%)*					
Proteinuric	59 (84)	n.a.	n.a.	n.a.	
Nephritic	11 (16)	n.a.	n.a.	n.a.	
First flare	41 (59)	n.a.	n.a.	n.a.	
Relapsing flare	29 (41)	n.a.	n.a.	n.a.	
*Renal Biopsy, n (%)*					
Class III	9 (13)	n.a.	n.a.	n.a.	
Class IV	53 (76)	n.a.	n.a.	n.a.	
Class V	8 (11)	n.a.	n.a.	n.a.	
Activity Index	7.5 ± 3.6	n.a.	n.a.	n.a.	
Chronicity Index	2.0 ± 1.9	n.a.	n.a.	n.a.	

Values are expressed as mean ± standard deviation (SD). BUN, blood urea nitrogen; eGFR, estimated glomerular filtration rate; anti-dsDNA, anti-double-stranded DNA (reference range <15 UI/mL); n.a., not applicable. ^a^
*p*-value refers to the comparison of the active lupus nephritis (LN) with healthy controls, the values in bracket referred to the comparison with active non-renal SLE and the values in square bracket referred to the comparison with the cohort of other glomerular-diseases: Mann-Whitney U Test or Pearson χ2 test.

**Table 2 ijms-20-04601-t002:** Baseline characteristics of patients according to clinical response to immunosuppressive therapy.

Characteristics	Responders		Non-Responders
	Active LN Cohort (*n* = 38)	Active LN Cohort (*n* = 32)	*p* Value
*Demographic*			
Age (years)	42 ± 12	39 ± 8	0.517
Gender (Female/male)	33/5	27/5	
*Race/ethnicity, n (%)*			
White	35 (92)	30 (94)	0.811
Hispanic Other	2 (5) 1 (3)	2 (6) 0	
*Laboratory parameters*			
Serum creatinine, mg/dL	1.0 ± 0.4	1.3 ± 0.6	0.025
eGFR (mL/min)	88 ± 26	75 ± 33	0.167
Urea (mg/dL)	48 ± 27	61 ± 30	0.054
Anti-dsDNA Abs, IU/mL	344 ± 84	220 ± 68	0.781
Serum C3, mg/dL	77 ± 32	79 ± 29	0.878
Serum C4, mg/dL	12.0 ± 9.6	13.7 ± 8.8	0.248
Proteinuria, g/24 h	3.0 ± 3.2	3.4 ± 2.7	0.563
Leukocytes (cel/µL)	95 ± 114	101 ± 190	0.495
Erythrocytes (cel/µL)	133 ± 131	187 ± 397	0.158
*Disease index (SLEDAI-2K)*			
Total score	14 ± 3	15 ± 2	0.145
Renal score	10 ± 2	9 ± 3	0.112
Extra-Renal score	5 ± 3	4 ± 2	0.184
*Renal Flare, n (%)*			
Proteinuric	38 (100)	14 (44)	<0.0001
Nephritic	0	18 (56)	<0.0001
First flare	27 (71)	14 (44)	0.021
Relapsing flare	11 (29)	18 (56)	0.021
*Renal Biopsy, n (%)*			
Class III	4 (11%)	3 (9%)	0.714
Class IV	29 (76%)	26 (82%)	0.618
Class V	5 (13%)	3 (9%)	0.441
Activity Index	7.4± 3.2	7.6 ± 4.0	0.864
Chronicity Index	1.7 ± 2.1	2.2 ± 1.8	0.203

Values are expressed by means ± standard deviation (SD). eGFR, estimated glomerular filtration rate; anti-dsDNA, anti-double-stranded DNA (reference range <15 UI/mL). *p*-value refers to the comparison of the Responders with Non-Responders in combined cohorts by Mann-Whitney U Test or Pearson χ2 test.

**Table 3 ijms-20-04601-t003:** Baseline characteristics of prospective cohort of patient according to clinical response to immunosuppressive therapy.

Characteristics	Responders (*n* = 22)	Non-Responders (*n* = 17)	*p* Value
*Demographic*			
Age (years)	42 ± 13	38 ± 9	0.198
Gender (Female/male)	17/5	8/9	
*Race/ethnicity, n (%)*			
White	21 (95)	14 (82)	0.193
Hispanics	1 (5)	3 (18)	
*Laboratory parameters*			
Serum creatinine, mg/dL	1.0 ± 0.7	1.3 ± 0.6	0.026
eGFR (mL/min)	88 ± 26	79 ± 33	0.172
Urea (mg/dL)	51 ± 27	61 ± 30	0.074
Anti-dsDNA Abs, IU/mL	324 ± 89	280 ± 85	0.819
Serum C3, mg/dL	75 ± 28	78 ± 30	0.941
Serum C4, mg/dL	12.4 ± 9.0	13.2 ± 8.7	0.637
Proteinuria, g/24 h	3.4 ± 3.1	3.3 ± 2.9	0.266
Leukocytes (cel/µL)	105 ± 114	101 ± 150	0.214
Erythrocytes (cel/µL)	183 ± 121	159 ± 197	0.431
*Disease index (SLEDAI-2K)*			
Total score	14 ± 3	15 ± 2	0.092
Renal score	10 ± 2	9 ± 3	0.249
Extra-Renal score	5 ± 3	4 ± 2	0.142
*Renal Flare, n (%)*			
Nephritic	0	11 (65)	0.082
Proteinuric	22 (100)	6 (35)	0.082
First episode	13 (59)	7 (41)	0.107
Relapsing	9 (41)	10 (59)	0.107
*Renal Biopsy, n (%)*			
Class III	3 (14)	2 (12)	0.212
Class IV	15 (68)	12 (71)	0.966
Class V	4 (18)	3 (18)	0.791
Activity Index	7.0± 3.4	7.4 ± 3.8	0.936
Chronicity Index	1.5 ± 2.0	2.0 ± 1.5	0.377

Values are expressed as mean ± standard deviation (SD). eGFR, estimated glomerular filtration rate; anti-dsDNA, anti-double-stranded DNA (reference range <15 UI/mL). *p*-value refers to the comparison of the Responders with NonResponders cohorts: Mann-Whitney U Test or Pearson χ2 test.

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
