# Peer review of "Urinary Neuropilin-1: A Predictive Biomarker for Renal Outcome in Lupus Nephritis"

_ijms, 2019, doi:10.3390/ijms20184601_

Round 1

Reviewer 1 Report

In their interesting study Maria Teresa Torres-Salido et al. analysed urine levels of neuropilin-1 (NRP-1) in two large cohorts of lupus nephritis (LN) patients (n=70, n=39). They showed higher levels of NRP-1 mRNA and protein in active LN as compared to active non-renal SLE. Urine NRP-1 levels were also initially higher in patients who did not properly respond to induction therapy (systemic [oral] methylprednisolone and oral MMF, 1-year observation). The authors further explored the role of NRP-1 receptor in primary renal endothelial and mesangial cells using CRISPR-Cas9 NRP1 gene knock-out, showing its contribution in promoting cell proliferation, migration (only in mesangium) and wound healing. Additionally, knock-out of NRP1 in peripheral blood T-cells resulted in their diminished response to VEGF-mediated suppression. However, the authors did not study in detail whether this phenomenon is similar both in pathogenic memory T-cells (e.g. CXCR3+ T-cells in case of LN) and in regulatory T-cells. The study is well designed and provides novel data on NRP-1/VEGF axis in LN. I have only few comments on possible change in data presentation, specifically to highlight the linkage of NRP-1 with the clinical presentation (nephritic syndrome) and its response to local inflammation (potential modulation by proinflammatory cytokines).

Major concerns

1.     Data presentation. In my opinion, the data from cohorts 1 and 2 should not be presented separately in the results section (but shown only combined as primary cohort). Alternatively, comparison of current cohort 1 and 2 can be moved to supplement. Thus, cohort 3 results might be used as a validation cohort. I think, this reflects better the study design and will improve clarity of the results.  

2.     Interpretation of results. Both in primary and validation (i.e. cohort 3) cohorts considerable fraction (~60%) of non-responding patients presented with nephritic syndrome, while such clinical presentation was not found in responder group. Higher urine levels of NRP-1 mRNA and protein could reflect the type of glomerular injury and inflammation. The authors should present additional data on potential linkage between NRP-1 and clinical and urinalysis features of the flare (e.g. if higher NRP-1 levels were restricted only to patients with overt nephritic syndrome). Did authors find any linkage of urine NRP-1 with renown urine inflammatory biomarkers of active LN (e.g. CCL2, CXCL10, CXCL16) That will also help to confirm whether urine NRP-1 indeed reflects pronounced (and potentially treatment resistant) renal inflammation.

Minor concerns

1.     Please provide information on number of replicates and statistical tests used in each figure legend. In some instances significance symbols (asterisk) do not show clearly which two groups differ. I propose using horizontal lines linking proper datasets with the significance symbol shown above.

2.     Figure 1 is cut at the border (I based the review on data presented in the text).

3.     Table 3 (Part: ‘renal flare’). Please check frequencies of first or relapsing flare as they don’t sum up to the number of patients in each group.

4.     The authors analyzed mRNA in urine sediment cells. Is there any linkage between urinary NRP1 expression and known urine transcriptome biomarkers of active LN including T-cell and podocyte signature (e.g. studies by Szeto CC and Klocke J). Is higher level of urine NRP-1 (in active LN) linked with increased renal expression (release) of this molecule in diseased glomeruli, or simply a presence of inflammatory cells (or damaged epithelial cells) in the urine? This is important from clinical point of view, as higher in situ production could herald smoldering inflammation and incoming renal flare in otherwise quiescent LN, before the rise of common urinalysis biomarkers and manifestation of clinical symptoms.

5.     In my opinion graphs showing long term changes in urine NRP-1 protein in the two responder groups (Fig. 3B) could be combined, two-way repeated model should be used to confirm significance of the change in NRP-1 levels. The same applies to VEGFA dataset.

6.     Assuming the linkage between renal inflammation and NRP-1, did Authors find any change in NRP-1 renal epithelial cell expression of NRP-1 in response to proinflammatory cytokines (i.e. linked with LN pathomechanisms).    

Author Response

REVIEWER 1:

In their interesting study Maria Teresa Torres-Salido et al. analysed urine levels of neuropilin-1 (NRP-1) in two large cohorts of lupus nephritis (LN) patients (n=70, n=39). They showed higher levels of NRP-1 mRNA and protein in active LN as compared to active non-renal SLE. Urine NRP-1 levels were also initially higher in patients who did not properly respond to induction therapy (systemic [oral] methylprednisolone and oral MMF, 1-year observation). The authors further explored the role of NRP-1 receptor in primary renal endothelial and mesangial cells using CRISPR-Cas9 NRP1gene knock-out, showing its contribution in promoting cell proliferation, migration (only in mesangium) and wound healing. Additionally, knock-out of NRP1 in peripheral blood T-cells resulted in their diminished response to VEGF-mediated suppression. However, the authors did not study in detail whether this phenomenon is similar both in pathogenic memory T-cells (e.g. CXCR3+ T-cells in case of LN) and in regulatory T-cells. The study is well designed and provides novel data on NRP-1/VEGF axis in LN. I have only few comments on possible change in data presentation, specifically to highlight the linkage of NRP-1 with the clinical presentation (nephritic syndrome) and its response to local inflammation (potential modulation by proinflammatory cytokines).

Major concerns

1. Data presentation. In my opinion, the data from cohorts 1 and 2 should not be presented separately in the results section (but shown only combined as primary cohort). Alternatively, comparison of current cohort 1 and 2 can be moved to supplement. Thus, cohort 3 results might be used as a validation cohort. I think, this reflects better the study design and will improve clarity of the results.

To better understand the paper and to continue preserving the original design of the study, we will present in the results section the baseline characteristics of the patients as a unique cohort, result of the combination of the two initial cohorts. Descriptions of Cohort 1 and cohort 2 are shown in the supplement (Page 2, line 71-75 and Table 1 and 2).

2. Interpretation of results. Both in primary and validation (i.e. cohort 3) cohorts considerable fraction (~60%) of non-responding patients presented with nephritic syndrome, while such clinical presentation was not found in responder group. Higher urine levels of NRP-1 mRNA and protein could reflect the type of glomerular injury and inflammation. The authors should present additional data on potential linkage between NRP-1 and clinical and urinalysis features of the flare (e.g. if higher NRP-1 levels were restricted only to patients with overt nephritic syndrome). Did authors find any linkage of urine NRP-1 with renown urine inflammatory biomarkers of active LN (e.g. CCL2, CXCL10, CXCL16) That will also help to confirm whether urine NRP-1 indeed reflects pronounced (and potentially treatment resistant) renal inflammation.

We correlated urinary NRP-1 levels with clinical, laboratory and immunological parameters. We did not find any correlation with the degree of proteinuria, estimated glomerular filtrated, histological type of nephritis, the level of creatinine, the presence of red blood cells or white blood cells in urine. No correlation with anti-dsDNA was found. It was found a correlation only with C4 levels (r=-0.241 and p= 0.045, graph was added in Figure 1c).

Initially, we did not correlate NRP-1 levels with proteinuric and nephritic flares. As suggested, we correlated both parameters and the analysis show an inverse correlation (r=-0.257, p=0.043). Therefore, higher NRP-1 levels slightly correlated with less nephritic flares, which were the ones associated with poor response. This information has been added in the manuscript (Page 3 line 103 and Figure 1c).

Currently, we are preparing another manuscript measuring simultaneously several urinary biomarkers to see their value to predict outcome in LN. Amongst them, there is CCL2 (MCP-1). For this reason, we decided not to include these results in this manuscript. However, after your suggestion, we show you that NRP-1 did not correlated with other urine inflammatory biomarkers of LN such as MCP-1 (Figure A) or NGAL (Figure B), that it is also a urinary biomarker of active LN (Torres-Salido MT et al. Neutrophil gelatinase-associated lipocalin as a biomarker for lupus nephritis. Nephrol Dial Transplant. 2014; 29:1740-9).

Minor concerns

1. Please provide information on number of replicates and statistical tests used in each figure legend. In some instances significance symbols (asterisk) do not show clearly which two groups differ. I propose using horizontal lines linking proper datasets with the significance symbol shown above.

We have provided the information in figure legends and we have modified figures using horizontal lines.

2. Figure 1 is cut at the border (I based the review on data presented in the text).
Figure 1 was not cut in our word format. We believe some problems in the vision of the figures might be the PDF form. We will make sure it does not happen again.

3. Table 3 (Part: ‘renal flare’). Please check frequencies of first or relapsing flare as they don’t sum up to the number of patients in each group.

Numbers have been corrected (Table 3)

3. The authors analyzed mRNA in urine sediment cells. Is there any linkage between urinary NRP1 expression and known urine transcriptome biomarkers of active LN including T-cell and podocyte signature (e.g. studies by Szeto CC and Klocke J).

Szeto CC et al published an increase of intra-renal expression of TWEAK and Fn14 in lupus patients (Lu J et al. Gene expression of TWEAK/Fn14 and IP-10/CXCR3 in glomerulus and tubulo-intrestitium of patients with lupus nephritis. Nephrology (Carlton). 2011; 16: 426-32). We have studied urinary TWEAK levels in our cohort, and we did not find difference between responder and non-responder LN patients. Urinary TWEAK levels related to the LN activity. Data will be published in another paper where it will be compared different urinary biomarkers.

Interesting, Klocke J et al compared urinary chemokines and urinary cell counts with lupus nephritis disease activity (Klocke J et al. Mapping urinary chemokines in human lupus nephritis: Potentially redundant pathways recruit CD4 and CD8 T cells and macrophages. Eur J Immunol. 2017; 47: 180-192). They found interindividual heterogeneity. We did flow cytometry analysis in urinary samples from a small cohort of LN patients (N=5 for each group, responder and non-responder). We were not able to detect sufficient amount of cell to be representative (less than 1000 events). We concentrated urine samples and repeated the experiment, but we were not successful. For this reason, we did not include these results that are not representative enough in the paper or discussion. Therefore, we did not have information to see if there is a linkage with T cell or podocyte signature.

Is higher level of urine NRP-1 (in active LN) linked with increased renal expression (release) of this molecule in diseased glomeruli, or simply a presence of inflammatory cells (or damaged epithelial cells) in the urine? This is important from clinical point of view, as higher in situ production could herald smoldering inflammation and incoming renal flare in otherwise quiescent LN, before the rise of common urinalysis biomarkers and manifestation of clinical symptoms.

Immunochemistry studies on renal tissue showed NRP-1 staining to e significantly increased in the LN group when compared to the control group. In addition, in those with CR the staining was found to be 3 times higher. Measurement of serum NRP-1 levels did not show differences between active lupus nephritis patients and controls. No differences were either found in responders and non-responders.

For this reason, we believe that there in an increased renal expression of this molecule. In vitro studies also support this hypothesis. Primary renal cell stimulation with inflammatory cytokines showed that NRP-1 was mainly produced by Mesangial cells following IL1α and by endothelial cells following VEGF.

5. In my opinion graphs showing long term changes in urine NRP-1 protein in the two responder groups (Fig. 3B) could be combined, two-way repeated model should be used to confirm significance of the change in NRP-1 levels. The same applies to VEGFA dataset.

Graphs have been changed as suggested (Figure 3b and 3c).

6. Assuming the linkage between renal inflammation and NRP-1, did Authors find any change in NRP-1 renal epithelial cell expression of NRP-1 in response to proinflammatory cytokines (i.e. linked with LN pathomechanisms).

To establish a link between inflammation and NRP-1 production, primary renal cells were stimulated by inflammatory cytokines such as IL1-α and VEGF. At 24 hours, following IL1-α stimulation, NRP-1 expression levels were significantly increased in mesangial renal cells when compared to the other cellular types (p<0.0001). However, following VEGF stimulation, endothelial renal cells exhibited the highest expression levels of NRP-1. Under stimulation, Tcells reduced significatively the NRP-1 expression.

Following stimulation, NRP-1 expression decreased significatively in tubular renal epithelial cells.

Data have been included in the text (Page 9, lines 228-233, Figure 4A)

Reviewer 2 Report

This manuscript demonstrated urinary Neuropilin-1 (NRP-1) as a biomarker to monitor the disease activity in patients with LN. Previous study (Vadasz et al., 2011) showed only phenotype such as increased renal expression of NRP1, whereas, this study demonstrated mechanistic roles of NRP-1 in LN pathogenesis, including angiogenesis effect on endothelial cells via VEGF and regeneration of the mesangium via PDGFB, and cytotoxicity of CD8+ T cells via VEGF.

Even a weak novelty and not deep, this manuscript was fairly described in detail. However, some should be revised.

Major revision

Authors divided patient groups into cohort 1, cohort 2, combined cohort. However, results according to combined or not were not different. Therefore, it had better to demonstrate only results of combined group. Most results demonstrated data according to response to immune therapy. Authors should describe criteria of responder and non-responder in methods section of main text body, not in supporting information. In page 9, lines 99-100, authors stated “Correlation analysis of uNRP-1 levels with other clinical and laboratory parameters only showed a correlation with C4 levels (r = - 0.182, p=0.008).” However, there is no figure or table to prove this result. There were no differences in urinary NRP-1 levels according to classes, which should be in depth discussed. Authors showed that urinary NRP-1 at baseline was high in responder, which can predict remission. If so, what can rheumatologist apply to patients with increased uNRP-1 level in Clinic field? Authors used urine pellet for measurement of urinary NRP-1 mRNA levels. Urinary NRP-1 mRNA levels in patients with active LN were increased. Which cells in urine pellets are supposed to produce NRP-1? What do this result mean?

        In addition, in Fig. 1, D is loss.

Author stated that NRP-1 increased cytotoxicity of CD8+ T cells via VEGF and reduce local inflammation. It is exaggerated that increased cytotoxicity of CD8+ T cells by NRP-1 can reduce local inflammation. Although renal CD8+ T cell infiltration was increased and correlated with renal activity, CD8+ T cells, mainly, don’t play an inflammatory role in pathogenesis. I recommend cytotoxicity of CD4+ T cells in fig S9 moves in fig 5 similarly with that of CD8+ T cells. Were levels of urinary NRP-1 correlated with those of kidney biopsy samples? Vasdaz et al showed that Semaphorine 3A staining was also observed predominantly in the renal tubuli, However, this study showed reduction of urinary samaphorine 3A in active nephritis. Why do authors think about discrepancy? Unnecessary results in supporting information can be mitigated. Authors didn’t need to display all data.

Minor revision

-What were kidney biopsy samples of control?

 It must be exactly mentioned in methods section.

-There are several typing errors including ‘Neprhitic’ (Table 1).

Author Response

REVIEWER 2:

This manuscript demonstrated urinary Neuropilin-1 (NRP-1) as a biomarker to monitor the disease activity in patients with LN. Previous study (Vadasz et al., 2011) showed only phenotype such as increased renal expression of NRP1, whereas, this study demonstrated mechanistic roles of NRP-1 in LN pathogenesis, including angiogenesis effect on endothelial cells via VEGF and regeneration of the mesangium via PDGFB, and cytotoxicity of CD8+ T cells via VEGF.
Even a weak novelty and not deep, this manuscript was fairly described in detail. However, some should be revised.

Major revision
Authors divided patient groups into cohort 1, cohort 2, combined cohort. However, results according to combined or not were not different. Therefore, it had better to demonstrate only results of combined group. Most results demonstrated data according to response to immune therapy. Authors should describe criteria of responder and non-responder in methods section of main text body, not in supporting information.
Criteria for responders and non-responders have been added to the main text (Page 15, line 394-399). As suggested, we will present only the combinatory cohort (Table 1 and 2)
In page 9, lines 99-100, authors stated “Correlation analysis of uNRP-1 levels with other clinical and laboratory parameters only showed a correlation with C4 levels (r = - 0.182, p=0.008).” However, there is no figure or table to prove this result. There were no differences in urinary NRP-1 levels according to classes, which should be in depth discussed.
We correlated urinary NRP-1 levels with clinical, laboratory and immunological parameters. We did not find any correlation with the degree of proteinuria, the level of creatinine, the presence of red blood cells or white blood cells in urine. No correlation with anti-dsDNA was found. We only find a correlation with C4 levels and we have added the graph in Figure 1C.
We described that amongst patients with LN, there were no differences in NRP-1 levels according to histological type or the degree of disease activity. However, this data needs to be taken with caution since approx 76% of patients had a type IV GMN. The other classes were not enough represented to draw conclusions (Page 3, lines 98-104).
Authors showed that urinary NRP-1 at baseline was high in responder, which can predict remission. If so, what can rheumatologist apply to patients with increased uNRP-1 level in Clinic field?
After these results, a further validation in a larger cohort of patients is required. We have done the sample size calculation required to validate the results and more than 200 patients with active LN would be required:
There is a project to enrol the to validate NRP-1 as a prognostic biomarker of LN response but the number of patients required makes it difficult.
If the results are validated, we believe NRP-1 would be a prognostic marker of clinical outcome. More interesting is the fact that NRP-1 may have a role in renal repair. This needs to be further studied.
Authors used urine pellet for measurement of urinary NRP-1 mRNA levels. Urinary NRP-1 mRNA levels in patients with active LN were increased. Which cells in urine pellets are supposed to produce NRP-1? What do this result mean?
Several studies have shown that the type of cells in the urine of patients with LN resembles the one observed in the kidney’s interstitial infiltration and consists mainly mainly of T cells and to a lesser extent of macrophages, B cells and plasmablasts/-cells Several authors describe a predominance of CD4+ T cells, while others report a majority of CD8+ T cells but also T reg had been described.
NRP1 expression has been characterized in different immune cellular phenotypes including macrophages, dendritic cells, and T cell subsets, especially regulatory T cell populations [1,2]. We performed stimulatory studies on several primary renal cells in vitro simulating an inflammatory environment in the kidney. Our in vitro studies only showed NRP-1 to be produced by mesangial cells after IL1α stimulation and by endothelial cells following VEGF. When all cellular types were stimulated simultaneously, no production of NRP-1 by T cells was observed. Therefore, we believe that NRP is produced locally in the kidney by mesangial cells and endothelial cells (Page 9, lines 227-233, Figure 4A).
1. The cellular signature of urinary immune cells in Lupus Nephritis: new insights into potential biomarkers. Kopetschke K, Klocke J et al. Arthritis Res Ther 2015; 3: 17:94.
2. Multifaceted Role of Neuropilins in the Immune System: Potential Targets for Immunotherapy. Sohini Roy, Arup K. Bag, Rakesh K. Singh, Surinder K. Batra and Kaustubh Datta. Front Immunol. 2017 Oct 10;8:1228
In addition, in Fig. 1, D is loss.
Our figures and tables in the word are correct. We assume that difficulties derive from the PDF creation.
Author stated that NRP-1 increased cytotoxicity of CD8+ T cells via VEGF and reduce local inflammation. It is exaggerated that increased cytotoxicity of CD8+ T cells by NRP-1 can reduce local inflammation. Although renal CD8+ T cell infiltration was increased and correlated with renal activity, CD8+ T cells, mainly, don’t play an inflammatory role in pathogenesis. I recommend cytotoxicity of CD4+ T cells in fig S9 moves in fig 5 similarly with that of CD8+ T cells.
It has been changed as your suggestion (Figure 5b).
Were levels of urinary NRP-1 correlated with those of kidney biopsy samples?
Urinary NRP-1 correlated significantly with renal NRP-1 in kidney biopsy (p=0.002, N=10). These results were added in the manuscript (Page 7, line 199).

Vasdaz et al showed that Semaphorine 3A staining was also observed predominantly in the renal tubuli, However, this study showed reduction of urinary samaphorine 3A in active nephritis. Why do authors think about discrepancy?
Vadasz et al showed an increased tubular Semaphorine 3A staining by immunohistochemistry and it was suggested that it could serve as a histological marker for tubular damage. The study found semaphorine 3 A to be inversely correlated with levels of proteinuria. They did not measure semaphorine 3A in urine. Our histological results also showed increased staining in LN, but mainly in non-responders in the tubuli thus, reinforcing its role in renal damage
In this study measurement by ELISA of serum or plasma levels in patients with SLE has found levels of semaphorine 3 A to be decreased in SLE patients compared with healthy controls. Levels were also inversely correlated with disease activity and mainly in those with nephritis.
Our study, the only one in the literature measuring semaphorine 3 A levels in urine in active LN patients found mRNA levels to be similar in patients with active LN and controls. However, at a protein level, measured by ELISA, semaphorine levels were found to be reduced in active LN compared with controls, but they could not distinguish responders from non-responders.
It is difficult to comment on discrepancies when there are no studies to compare in urine. The decreased levels in urine could mirror the levels describe in serum and be correlated inversely with disease activity. In other renal pathologies urinary, in some models it has been found an early biomarker of disease which high levels disappear over time. Further studies are required to be able to compare
Unnecessary results in supporting information can be mitigated. Authors didn’t need to display all data.
We have eliminated some repeated graphs from supporting information (Figure S1).

Minor revision
-What were kidney biopsy samples of control? It must be exactly mentioned in methods section.
As controls, we used disease-free kidney sections from tissue margins of total or subtotal nephrectomies obtained from patients undergoing surgery for renal malignancies. It has been moved from SI to the main manuscript (Page 15, line 435-436).

-There are several typing errors including ‘Neprhitic’ (Table 1).
It has been corrected (Table 1).

Reviewer 3 Report

Overall comments:

In the current study by Toress-Salido et.al., authors evaluated a significant elevation of urinary neuropilin-1 (uNRP-1), a receptor for several members of the vascular endothelial growth factor (VEGF) family, in SLE patients with renal involvement. Authors’ findings that patients with increased uNRP-1 presented favorable clinical outcome in response to the treatment is very interesting. In addition, their analysis to compare NRP-1, VEGFA and VEGF receptors in active lupus urine samples further emphasizes the clinical significance of uNRP-1 as the potential biomarkers to predict patient prognosis. Moreover, in vitro data to demonstrated molecular functions of NRP-1 on human endothelial cells and mesangial cells efficiently support their clinical findings. Taken together, their study seemed to have value for publication, but authors need to improve their data presentation for appropriate review.

Major comments:

Page 3, Results, 2.2. Urinary expression of NRP-1 in Lupus Nephritis; Authors presented NRP-1 mRNA levels by real-time PCR in urine sample. According to the methods section, I understood that cells collected in centrifuged urine samples were subjected to PCR analysis. How they purified cells from pellets in centrifuged urine samples? or they extracted mRNA from whole pellets that include cells, debris, exosomes, and filtrated mRNA? When they subjected whole pellets, GAPDH is not available for internal control to standardize NRP-1 mRNA. Authors described that 41 (58.6%) patients presented the first renal episode. But the number of patients already received the treatment with steroid or other immunesuppressants are not described. Because the treatment for renal or other organ involvements of SLE may impact on uNRP-1 levels, authors need compare uNRP-1 levels between non-treated and treated patients. Because legends for all supplementary figures are lacking in the main text or below supplemental figures, the review was not done.

Minor comments:

All figures; Capital alphabets are used for panels in figures, but those are not capitalized in legends. Is this the journal style? Page 2, line 74; “GMN” should be spelled out at the first appearance. Page 3, Table 1, mg/D after serum creatinine should be corrected universal unit such as mg/dL. Page 5, Figure 1.; Data for “D” is totally missing in the file. Page 8, Figure 2.; ROC of uNRP-1 in A is almost invisible. ROCs of uNRP-1 in A and uVEGFA in Figure S5 are repeatedly appeared in E. Authors should present ROCs of NRP-1/Cr, VEGF/Cr, Serum Cr, and proteinuria in A, and should delete E and Figure S5. Page 8, Figure 2B.; Kaplan-Myer curve in B is almost invisible. Page 8, Figure 2E.; NRP-1/Cr and VEGF/Cr should be corrected to uNRP-1/uCr and uVEGFA/uCr, respectively. Page 9, Figure S3.; unit for VEGFA is presented as pg/mg Cr in Y axis, but “Cr” after ng/ml are missing in SEMA3A, VEGFR1, and VEGFR2.

Author Response

REVIEWER 3:

Overall comments:

In the current study by Toress-Salido et.al., authors evaluated a significant elevation of urinary neuropilin-1 (uNRP-1), a receptor for several members of the vascular endothelial growth factor (VEGF) family, in SLE patients with renal involvement. Authors’ findings that patients with increased uNRP-1 presented favorable clinical outcome in response to the treatment is very interesting. In addition, their analysis to compare NRP-1, VEGFA and VEGF receptors in active lupus urine samples further emphasizes the clinical significance of uNRP-1 as the potential biomarkers to predict patient prognosis. Moreover, in vitro data to demonstrated molecular functions of NRP-1 on human endothelial cells and mesangial cells efficiently support their clinical findings. Taken together, their study seemed to have value for publication, but authors need to improve their data presentation for appropriate review.

Major comments:

Page 3, Results, 2.2. Urinary expression of NRP-1 in Lupus Nephritis; Authors presented NRP-1 mRNA levels by real-time PCR in urine sample. According to the methods section, I understood that cells collected in centrifuged urine samples were subjected to PCR analysis. How they purified cells from pellets in centrifuged urine samples? or they extracted mRNA from whole pellets that include cells, debris, exosomes, and filtrated mRNA? When they subjected whole pellets, GAPDH is not available for internal control to standardize NRP-1 mRNA.

We have extracted mRNA from urinary pellet cell, we did not purified urinary cells from patients samples. Urinary exosomes are not present in urinary pellet. To obtain them by normal centrifugation you should use specific kit extraction or you should use ultracentrifugations (Solé C et al. miR-29c in urinary exosomes as predictor of early renal fibrosis in lupus nephritis. Nephrol Dial Transplant. 2015; 30:1488-96). Debris and mRNA filtrated could be presented in urinary pellet; for this reason, we have evaluated NRP-1 levels in serum and also in renal kidney biopsies to evaluate NRP1 renal production (Page 3, line 104, Figue S1 and Figure 3). To study deepdly NRP1 renal production, we have included more in vitro experiments in result section (Page 9, lines 228-233).

We used GAPDH to normalizate expression levels by a reference gene. Normalization by a refrence gene is necessary to supress the effect of RNA degradation. It is a robust, easy and widely used technique but it is depent on the quality of the reference gene. In the literatura, it has been described that GAPDH are prefereable reference genes for urinary mRNA than 18S or HPRT (Galichon P et al. Urinary mRNA for the Diagnosis of Renal Allograft Rejection: The Issue of Normalization. Am J Transplant. 2016; 16:3033-3040). In order to increase the quality of the paper, we have determinate the relative expression between groups using 18S as reference gene in the place of GAPDH. Similar results were found between the two reference genes (r=0.9686 , p<0.001):

This result has not been included in the paper because it has not been provided new insights.

Authors described that 41 (58.6%) patients presented the first renal episode. But the number of patients already received the treatment with steroid or other immunesuppressants are not described. Because the treatment for renal or other organ involvements of SLE may impact on uNRP-1 levels, authors need compare uNRP-1 levels between non-treated and treated patients. Because legends for all supplementary figures are lacking in the main text or below supplemental figures, the review was not done.

 There were non untreated patients. Patients with first renal flare were patients with an already known SLE (mean disease duration 3±6 years) receiving antimalarials, low dose prednisone or inmunosupressive agents such as azathioprine or MTX. Those with a previous episode were also receiving similar treatment with antimalarials, low dose prednisone and low dose mycophenolate (25%) or no inmunosuppressive agents (16.4%).

Levels of urinary NRP-1 were compared between first flare and relapsing condition and no differences were found as a group or comparing responders and non-repsonders.

Minor comments:

All figures; Capital alphabets are used for panels in figures, but those are not capitalized in legends. Is this the journal style? Page 2, line 74; “GMN” should be spelled out at the first appearance. Page 3, Table 1, mg/D after serum creatinine should be corrected universal unit such as mg/dL. Page 5, Figure 1.; Data for “D” is totally missing in the file. Page 8, Figure 2.; ROC of uNRP-1 in A is almost invisible. ROCs of uNRP-1 in A and uVEGFA in Figure S5 are repeatedly appeared in E. Authors should present ROCs of NRP-1/Cr, VEGF/Cr, Serum Cr, and proteinuria in A, and should delete E and Figure S5. Page 8, Figure 2B.; Kaplan-Myer curve in B is almost invisible. Page 8, Figure 2E.; NRP-1/Cr and VEGF/Cr should be corrected to uNRP-1/uCr and uVEGFA/uCr, respectively. Page 9, Figure S3.; unit for VEGFA is presented as pg/mg Cr in Y axis, but “Cr” after ng/ml are missing in SEMA3A, VEGFR1, and VEGFR2. 

As suggested, they have been corrected.

Reviewer 4 Report

Torres-Salido, Sacshis, and Sole et al. conducted cross-sectional and longitudinal prospective studies, and in vitro experiments to show how urinary NRP1 (uNRP1) and urinary VEGFA (uVEGFA) affect pathogenesis of lupus nephritis (LN). In the cross-sectional study, urinary mRNA and protein expression of NRP1 and VEGFA were proved to be significantly higher in active LN than SLE without LN and healthy control, and they were similarly proved to be significantly higher in responders than those in non-responders. Correlation of urinary protein levels of RNP1 and VEGFA was statistically significant, although uVEGFA protein profile to predict remission was inferior to that of uRNP1. Urinary SEMA3A levels were significantly less in active LN than those of SLE without LN and healthy control. Histological protein expression of NRP1 and VEGFA was also tested, and NRP1 protein was significantly expressed in kidney tissue in responders more than that in non-responders. In the prospective study, uNRP1 protein levels were shown to decrease gradually after treatment, but interestingly, they were higher in responders than those in non-responders at all time points as Flare, 3th, 6th, 9th, and 12th month. uVEGFA protein levels were also shown to decrease significantly, and they were higher in responders than those in non-responders at Flare, and 3th month, going back to equal levels of non-responders afterwards. In vitro experiments using HREC cells, HRMC cells and T cell from LN patients, NRP1 was shown to have angiogenesis effects via VEGF, to promote mesangial cell migration via PDGFB, and to induce VEGF-dependent T cell cytotoxicity.

General comments

With large various data from clinical and basic studies, authors clarified NRP1 roles in the pathogenesis of LN. Furthermore, this study showed the possibility of uNRP1 to serve as the predictive marker of treatment response in LN for the first time. From that point of view, this manuscript is precious, and deserve to be publication. However, there are so many various errors which need correction before publication, and that makes disappointingly this manuscript detract from quality. The reasons of these so many errors seemed to be caused by the careless editing process. Authors should take more attention to improve the quality of manuscript before submission.  

Major comments

Please show whole data of Figure 1 on page 5. Part of Figure 1 was cut and reviewers could not see all data.

Graphic description of Figure 2 A and B were unclear.

Describe the detailed definition of active LN and remission more clearly in the Materials and Methods part. For instance, ‘Active LN was defined as…’ , and please use the same term and same definition all through this manuscript. Similarly, ‘response’ and ‘remission’ are confusing. Please explain each definition more clearly. If they are same, please use the same term all through the manuscript.

In Table 1, please show correct data regarding gender in cohort 2.

In page 6, the last line, discuss the reason of the reduced protein levels of SEMA3A in the discussion part.

In page 7, line 16, please explain how authors concluded that the protein expression of NRP1 in tubuli was less than glomeruli without quantitative data.

In Table 3, there are so much inconsistency in the data in gender and Renal Flare. Some of the data are more than total number.

Please discuss the reason why the protein levels of VEGFA were not significantly different between responders and non-responders, although the urinary protein levels of VEGFA seemed to be significantly higher in responders than those in non-responders during the follow-up of first 6 months.

The description of immunohistochemical scoring system is insufficient. Please improve the description for readers to understand how the positive cell percentage score and intensity score were integrated into the average score.

Same comment for the description of the scoring system of immunocytochemistry in SI Materials and Methods on page 5. Besides, please change the headline of this part from Evaluation of immunofluorescence and immunohistochemistry to Evaluation of immunocytochemistry.

In reference of SI Materials and Methods, please refer to original publication rather than recommendation paper by Betsias et al.

For statistics, please use ANOVA and posttest to test the significance of difference, if protein levels of a certain target molecule would be compared among over three groups, and between two groups out of all.

Minor comments

On Page 2, 2. Results, 2.1. Patient part. Please show what GMN means, because this is first appearance in the manuscript for this abbreviation.

Figure legend of Figure 1. Please include the term ‘NRP1’.

In Figure 3 legend, please move ‘(c)’ to after ‘(b)’. In all figure legends, please use same large letters to depict A, B, C…. for each part of figure.

In reference of SI Materials and Methods, please correct the referred publication.

For instance, there is no reference numbered as 40 listed in the reference, although [40] was used in the page 3. Furthermore, same in page 3, ISN/RPS classification should be referred to [37] instead of [38].

Author Response

REVIEWER 4:

 Torres-Salido, Sacshis, and Sole et al. conducted cross-sectional and longitudinal prospective studies, and in vitro experiments to show how urinary NRP1 (uNRP1) and urinary VEGFA (uVEGFA) affect pathogenesis of lupus nephritis (LN). In the cross-sectional study, urinary mRNA and protein expression of NRP1 and VEGFA were proved to be significantly higher in active LN than SLE without LN and healthy control, and they were similarly proved to be significantly higher in responders than those in non-responders. Correlation of urinary protein levels of RNP1 and VEGFA was statistically significant, although uVEGFA protein profile to predict remission was inferior to that of uRNP1. Urinary SEMA3A levels were significantly less in active LN than those of SLE without LN and healthy control. Histological protein expression of NRP1 and VEGFA was also tested, and NRP1 protein was significantly expressed in kidney tissue in responders more than that in non-responders. In the prospective study, uNRP1 protein levels were shown to decrease gradually after treatment, but interestingly, they were higher in responders than those in non-responders at all time points as Flare, 3th, 6th, 9th, and 12th month. uVEGFA protein levels were also shown to decrease significantly, and they were higher in responders than those in non-responders at Flare, and 3th month, going back to equal levels of non-responders afterwards. In vitro experiments using HREC cells, HRMC cells and T cell from LN patients, NRP1 was shown to have angiogenesis effects via VEGF, to promote mesangial cell migration via PDGFB, and to induce VEGF-dependent T cell cytotoxicity.

  General comments

 With large various data from clinical and basic studies, authors clarified NRP1 roles in the pathogenesis of LN. Furthermore, this study showed the possibility of uNRP1 to serve as the predictive marker of treatment response in LN for the first time. From that point of view, this manuscript is precious, and deserve to be publication. However, there are so many various errors which need correction before publication, and that makes disappointingly this manuscript detract from quality. The reasons of these so many errors seemed to be caused by the careless editing process. Authors should take more attention to improve the quality of manuscript before submission.  

Major comments

 Please show whole data of Figure 1 on page 5. Part of Figure 1 was cut and reviewers could not see all data.

We have added an improved Figure.

 Graphic description of Figure 2 A and B were unclear.

We have improved it.

 Describe the detailed definition of active LN and remission more clearly in the Materials and Methods part. For instance, ‘Active LN was defined as…’ , and please use the same term and same definition all through this manuscript. Similarly, ‘response’ and ‘remission’ are confusing. Please explain each definition more clearly. If they are same, please use the same term all through the manuscript.

We have added the definition of active LN in the material and methods of the main manuscript (page 15, line 407-410).

We have corrected in all the manuscript and maintained the same definition. Active LN and responders and non-responders.

 In Table 1, please show correct data regarding gender in cohort 2.

We have corrected the data (Table 1).

 In page 6, the last line, discuss the reason of the reduced protein levels of SEMA3A in the discussion part.

Vadasz et al showed an increased tubular Semaphorin 3A staining by immunohistochemistry and it was suggested that it could serve as a histological marker for tubular damage. The study found semaphorin 3 A to be inversely correlated with levels of proteinuria. They did not measure semaphorin 3A in urine. Our histological results also showed increased staining in LN, but mainly in non-responders in the tubuli thus, reinforcing its role in renal damage.

Measurement by ELISA of serum or plasma levels in patients with SLE has found levels of semaphorin 3 A to be decreased in SLE patients compared with healthy controls. Levels were also inversely correlated with disease activity and mainly in those with nephritis.

Our study, the only one in the literature measuring semaphorine 3 A levels in urine in active LN patients found mRNA levels to be similar in patients with active LN and controls. However, at a protein level, measured by ELISA, semaphorin levels were found to be reduced in active LN compared ot controls, but they could not disintinguish responders from non-responders.

It is difficult to comment on discrepancies when there are no studies to compare in urine. The decreased levels in urine could mirror the levels describe in serum and be correlated inversely with disease activity. In other renal pathologies urinary, in some models it has been found an early biomarker of disease which high levels disappear over time. Further studies are required to be able to compare.

 In page 7, line 16, please explain how authors concluded that the protein expression of NRP1 in tubuli was less than glomeruli without quantitative data.

NRP-1 staining in LN renal biopsies was localized in mesangium and also in the tubular structure. However, the main difference in staining between responder and non-responder was a higher NRP-1 expression in the glomeruli in the responder group compared to non-responders.

It has been corrected and added in the Figure 3A and in the text (Page 7, lines 195-197).

 In Table 3, there are so much inconsistency in the data in gender and Renal Flare. Some of the data are more than total number.

Data had been corrected

 Please discuss the reason why the protein levels of VEGFA were not significantly different between responders and non-responders, although the urinary protein levels of VEGFA seemed to be significantly higher in responders than those in non-responders during the follow-up of first 6 months.

In agreement with literature, we found that patients with active LN had increased VEGF levels in renal tissue compared to controls (Frieri M et al. Toll-like receptor 9 and vascular endothelial growth factor levels in human kidneys from lupus nephritis patients. J Nephrol 2012; 25:1041-6.). However, we did not found significant differences in VEGFA renal tissue between responders and non-responders group. In contrast, we observed high levels of urinary VEGFA protein in responder patients. It has also been confirmed in the six first months during the follow-up. In a similar way is observed this tendency with SEMA3A levels and only high NRP-1 levels were clearly observed in kidney and urinary samples. We do not know exactly the reason of it, maybe could be that immunoshistochemistry is not enough sensible to detect small differences or maybe could be for the low number of samples (N= 5 for renal immunohistochemistry). However, we have included a small discussion about it in the manuscrit (Page 13, lines 31-320).

The description of immunohistochemical scoring system is insufficient. Please improve the description for readers to understand how the positive cell percentage score and intensity score were integrated into the average score.

Results were evaluated on blinded specimens by the Vall d’Hebrón pathologist unit under the supervision of the nephropathologist (Dr.Marta Vidal). The percentage of cells expressing the different probes was scored semiquantitatively as follows: 0 (no expression), 1 (11-20%), 2 (40-60%), or 3 (>80%). Staining intensity was scored semiquantitatively as 0 (no staining), 1 (weakly positive), 2 (moderately positive), or 3 (strongly positive). This scores were obtained after the evaluation of 5 pathologists. After that, the mean of them has been expressed in the figures as “Average score”. This explanation has been added in SI Methods (“Evaluation of immunohistochemistry”).

Same comment for the description of the scoring system of immunocytochemistry in SI Materials and Methods on page 5. Besides, please change the headline of this part from Evaluation of immunofluorescence and immunohistochemistry to Evaluation of immunocytochemistry.

As suggested, it has been corrected.

 In reference of SI Materials and Methods, please refer to original publication rather than recommendation paper by Betsias et al.

It has been changed for the original publication.

 For statistics, please use ANOVA and posttest to test the significance of difference, if protein levels of a certain target molecule would be compared among over three groups, and between two groups out of all.

The one-way analysis of variance (ANOVA) is employed to compare the means of three or more independent data sets that are normally distributed. Multiple measurements from the same set of subjects cannot be treated as separate, unrelated data sets. Comparison of means in such a situation requires repeated measures ANOVA. It is to be noted that while a multiple group comparison test such as ANOVA can point to a significant difference, it does not identify exactly between which two groups the difference lies. To do this, multiple group comparison needs to be followed up by an appropriate post hoc test.

In our project, continuous variables were compared using Student’s t-test or one-way ANOVA followed by Bonferroni test when appropriate. Categorical data were compared by the X2 test. Correlations were determined by Spearman or Pearson correlation when appropriate. It has been explained better in methodology (Stadistic analysis, Page 17, 492-495 lines) and added in each figure legend.

Minor comments

 On Page 2, 2. Results, 2.1. Patient part. Please show what GMN means, because this is first appearance in the manuscript for this abbreviation.

It has been added.

 Figure legend of Figure 1. Please include the term ‘NRP1’.

It has been added.

 In Figure 3 legend, please move ‘(c)’ to after ‘(b)’. In all figure legends, please use same large letters to depict A, B, C…. for each part of figure.

As suggestion, it has been changed. We have corrected figure legends following journal style.

 In reference of SI Materials and Methods, please correct the referred publication.

It has been corrected.

For instance, there is no reference numbered as 40 listed in the reference, although [40] was used in the page 3. Furthermore, same in page 3, ISN/RPS classification should be referred to [37] instead of [38].

References have been revised.

Round 2

Reviewer 2 Report

This manuscript was revised properly according to reviewer's comments.

Reviewer 3 Report

Because the manuscript has been significantly improved, I have no concerns for the publication.

Reviewer 4 Report

Authors properly responded to all major and minor comments.